# Biodiversity among *Brettanomyces bruxellensis* Strains Isolated from Different Wine Regions of Chile: Key Factors Revealed about Its Tolerance to Sulphite

**DOI:** 10.3390/microorganisms8040557

**Published:** 2020-04-13

**Authors:** Camila G-Poblete, Irina Charlot Peña-Moreno, Marcos Antonio de Morais, Sandra Moreira, María Angélica Ganga

**Affiliations:** 1Departamento de Ciencia y Tecnología de los Alimentos, Facultad Tecnológica, Universidad de Santiago de Chile, Santiago 9170020, Chile; camila.gonzalezpo@usach.cl (C.G.-P.); sandra.moreira.r@gmail.com (S.M.); 2Department of Genetics, Biosciences Center, Universidade Federal de Pernambuco, Recife 50670-901, Brazil; irina.penamoreno@gmail.com (I.C.P.-M.); marcos.moraisjr@ufpe.br (M.A.d.M.J.)

**Keywords:** *Brettanomyces bruxellensis*, sulphur dioxide, *p*-coumaric acid, spoilage microorganisms

## Abstract

*Brettanomyces bruxellensis* is regarded as the main spoilage microorganism in the wine industry, owing to its production of off-flavours. It is difficult to eradicate owing to its high tolerance of adverse environmental conditions, such as low nutrient availability, low pH, and high levels of ethanol and SO_2_. In this study, the production of volatile phenols and the growth kinetics of isolates from various regions of Chile were evaluated under stressful conditions. Through randomly amplified polymorphic DNA (RAPD) analysis, 15 strains were identified. These were grown in the presence of *p*-coumaric acid, a natural antimicrobial and the main precursor of off-flavours, and molecular sulfur dioxide (mSO_2_), an antimicrobial synthetic used in the wine industry. When both compounds were used simultaneously, there were clear signs of an improvement in the fitness of most of the isolates, which showed an antagonistic interaction in which *p*-coumaric acid mitigates the effects of SO_2_. Fourteen strains were able to produce 4-vinylphenol, which showed signs of phenylacrylic acid decarboxylase activity, and most of them produced 4-ethylphenol as a result of active vinylphenol reductase. These results demonstrate for the first time the serious implications of using *p*-coumaric acid, not only for the production of off-flavours, but also for its protective action against the toxic effects of SO_2_.

## 1. Introduction

Yeasts belonging to the *Brettanomyces* genus are generally regarded as the main spoilage microorganisms in the wine industry as their presence alters the organoleptic characteristics of the wine, and thus leads to significant economic losses throughout the world [1]. *Brettanomyces bruxellensis*, also known as the teleomorph *Dekkera bruxellensis*, is one of the species recognized in the wine industry for conferring unwanted olfactory characteristics similar to wet wool, musty and medicinal odours, burned plastic or wet horsehair (Brett characters) through the production of volatile off-flavours in the winemaking process [2]. These particular aromas are produced from phenolic compounds that are present in the grape and musts that are called hydroxycinnamic acids (AHC) [3]. The compounds, which include *p*-coumaric, ferulic and caffeic acids, are endogenous components of grapes and have been described as preservatives of natural foods [4] because of their antimicrobial activity [5,6]. *Brettanomyces bruxellensis* is able to metabolize AHCs during the winemaking by means of a phenylacrylic acid decarboxylase [7], and convert them to hydroxystyrenes (vinylphenols), which are then reduced to ethyl derivatives by a NADH-dependent vinyl phenol reductase to produce ethylphenols [8,9].

Due to its harmful effects on wine quality, the elimination of *B. bruxellensis* from the fermentation processes is very important. However, this has proved to be a difficult task on account of its tolerance to adverse environmental conditions such as low nutrient availability, low pH and high levels of ethanol [10,11,12]. However, there are several techniques that can be employed to restrict or prevent the growth of this yeast in the wine, such as the addition of sulphur dioxide (SO_2_), in the form of potassium metabisulphite (PMB), which is the chemical antimicrobial agent that is most widely used in the control of unwanted microorganisms [13]. Additionally, molecular SO_2_ (mSO_2_) is an oxidizing agent that is used in winemaking for controlling and stabilizing the end product. Sulphurous anhydride is generally added to musts and wines as an aqueous solution in concentrations ranging from 0.3 to 0.8 mg L^−1^ in the red wine technology [14]. SO_2_ in sufficient inhibitory concentration is capable of inhibiting enzymes such as glyceraldehyde-3-phosphate dehydrogenase, ATPase, alcohol dehydrogenase, aldehyde dehydrogenase and NAD^+^-dependent glutamate dehydrogenase, which can affect key metabolic processes and lead to cell death [15]. Despite this, a high SO_2_ concentration can lead to altered sensory characteristics of the wine. However, numerous authors have stated that the use of increasing concentrations should take account of the specific physiological response to the genetic constitution of *B. bruxellensis* strains and the tolerance of these strains to this agent [12]. This is a very serious matter as wine-producing regions can harbour strains of different clonal origins that, as a result of variations in genetic constitution and metabolic profiling, can vary in their levels of tolerance, as well as their capacity to produce off-flavours. Chile is an important wine producer in the world, and its grapes are grown in several regions and fermented in different winemaking conditions. In light of this, the aim of this study was to investigate the genetic diversity, the physiological characteristics and the growth fitness in the presence of the combination of the antimicrobials SO_2_ and *p*-coumaric acid of 15 isolates of *B. bruxellensis* collected from fermentation processes in various regions of Chile. The evaluation of the yeast response to the inhibitors allowed understanding of the complexity of the yeast resistance and their influence on the production of aromatic compounds.

## 2. Materials and Methods

### 2.1. Strain Selection and Cell Maintenance

The strains stored in the Laboratory of Biotechnology and Applied Microbiology of the University of Santiago de Chile (LAMAP) were used in this study. The isolated strains were selected from wine fermentation processes from several regions in Chile. Initially the cells were activated in selective medium for *B. bruxellensis*, corresponding to solid YPD with calcium carbonate (yeast extract 10 g L^−1^, peptone 20 g L^−1^, glucose 20 g L^−1^, agar 20 g L^−1^, CaCO_3_ 5 g L^−1^) [7]. The cells were incubated in Petri dishes for 6–10 days at 28 °C.

The origin of *B. bruxellensis* strains correspond to: L-2472, L-2474, L-2476, L-2480 and L-2478 from Alto Jahuel (33°44′01″ S; 70°41′03″ O), L-2570, L-2482, L-2755 and L-2597 from Rengo (34°24′23″ S; 70°51′30″ O), L-2676, L-2679 and L-2690 from Molina (35°21′12.13″ S; 70°54′34.34″ O) and L-2731, L2742, L-2759 and L-2763 from Nancagua (34°40′3.94″ S; 71°11′30.98″ O).

### 2.2. Molecular Identification

Yeast cells were cultivated in synthetic medium consisting of 2% glucose and 6.7 g L^−1^ yeast nitrogen base (YNB) (Difco Laboratories, Detroit, USA) and distilled water to pH 6.0 [7]. These assays were done under constant agitation (120 r.p.m.) at 28 °C for 7 days (aerobic condition). Genomic DNA extraction was performed using Wizard^®^ Genomic DNA Purification Kit with modifications (Promega, WI, USA).

The extracted DNA was analysed by PCR amplification using ITS1 (5 ′TCCGTAGGTGAACCTGCGG 3′) and ITS4 (5 ′TCCTCCGCTTATTGATATGC3 ′) primers [16]. The reaction mixture contained 1× buffer ABM, 1.5 mM MgSO_4_, 0.2 mM of each dNTP, 0.5 μM of each primer, 1.25 U Taq DNA polymerase (ABM, Richmond, VA, Canada) and 80 ng of DNA template. Amplification reactions were carried out in a Peltier Thermal Cycler (PT−100) under the following conditions: denaturation at 94 °C for 5 min followed by 30 cycles of amplification with denaturation of 94 °C for 1 min, annealing at 55 °C for 1 min and extension of 72 °C for 2 min, with a final extension at 71 °C for 10 min. PCR products were visualized by means of electrophoresis using 1.5% agarose gel at 90 V for 60 min in 0.5× TBE buffer.

The RAPD-PCR technique was used for clonal discrimination between the yeast isolates. Three primers from the OPA series were used: OPAE09 (5′-TGCCACGAGG-3′), OPAD08 (5′-GGCAGGCAAG-3′) and OPAE12 (5′-CCGAGCAATC-3′) (Operon Technologies, CA, USA), which were previously selected in LAMAP-USACH as being suitable for *B. bruxellensis* strain discrimination (unpublished data). PCR was performed using a final volume of 20 µL, containing 1× buffer ABM, 1.5 mM MgSO_4_, 0.2 mM of each dNTP, 0.4 μM of each primer, 1.25 U Taq DNA polymerase (ABM) and 100 ng of DNA template. The amplification reactions consisted of initial denaturation at 94 °C for 2 min followed by 25 cycles of amplification with denaturation of 94 °C for 1 min, annealing at 35 °C for 45 s and extension of 72 °C for 1 min, with a final extension at 71 °C for 10 min. The amplicons were subjected to electrophoresis in 5% polyacrylamide gel matrix (5% acrylamide-bisacrylamide 3008 in TBE 0.5× containing 20% ammonium persulfate and 50% tetramethylethylenediamine (TEMED)) at 70 volts for 130 min. The size of the bands was detected by using Quantity One software (Bio-Rad, CA, USA).

### 2.3. Growth Kinetics Using Micro-Cultures in the Presence of p-Coumaric Acid (APC) and Sulphur Dioxide (SO_2_)

Yeast cells were activated by inoculation in 5 mL of liquid YPD minimum medium (1% yeast extract, 2% glucose and 2% peptone) for 8 days at 28 °C with constant agitation [17]. Subsequently, 100 µL of each culture was transferred to 5 mL of synthetic medium and incubated for 8 days at 28 °C. These were used as seed cultures for the tests in the presence of inhibitors. This involved diluting seed cells in synthetic media to 5 × 10^5^ cells mL^−1^ to 200 µL final volume. Six different conditions were tested: (a) medium without inhibitor (control), (b) medium containing potassium metabisulfite (mSO_2_) as a producer of sulphur dioxide at 0.3 mg L^−1^, (c) medium containing potassium metabisulfite (mSO_2_) as a producer of sulphur dioxide at 0.6 mg L^−1^, (d) medium containing *p*-coumaric acid at 100 mg L^−1^, (e) medium containing 100 mg L^−1^ of *p*-coumaric acid and sulphur dioxide at 0.3 mg L^−1^, (f) medium containing 100 mg L^−1^ of *p*-coumaric acid and sulphur dioxide at 0.6 mg L^−1^. The pH of the culture media was adjusted to 3.5 with HCl 1 M (average value for pH generally encountered in winemaking conditions) [18]. The kinetics of the cell growth of each strain in these conditions was evaluated by means of growth curves and using sterile microplates containing 96 wells, with absorbance measurements made every 30 min in a TECAN microplate reader. Absorbance measurements were taken from the microplates (OD 600 nm × time) [18] with agitation at 120 (r.p.m) before each reading, in biological triplicate and technical triplicate. The experiments were carried out for 10–25 days at 28 °C.

### 2.4. Quantification of Volatile Phenols

Volatile phenol concentration was quantified using the method described by [19]. For this, we utilized the HPLC technique (Shimadzu Corporation, Kyoto, Japan). The column used was C18 Shimadzu reverse phase (150 × 4.6 mm). The solvent system corresponded to a water/formic acid gradient (90:10% *v*/*v*) with methanol. Standard curves were prepared for *p*-coumaric, 4-vinilphenol and 4-etilphenol (Sigma Aldrich, St. Louis, MO, USA) in the range of 0.5–100 mg L^−1^.

All of the strains were cultivated in synthetic medium (YNB) supplemented with 100 mg L^−1^
*p*-coumaric acid. Two milliliters of the culture media of all strains, which reached an OD 600 nm of 0.8, was centrifuged at 16,000× *g* for 10 min at 20 °C. The supernatant was collected and refrigerated at 4 °C until the analysis.

### 2.5. Principal Component Analysis and Clustering

Relative values regarding the growth rate and lag phase were calculated for each treatment condition used. This was achieved by dividing the absolute value of the physiological parameterwhen the cells were cultivated in one of the treatment conditions (combinatory effect of *p*-coumaric acid and mSO_2_)—by the absolute value in the reference condition. Consumption of *p*-coumaric acid was defined as the percentage of the acid consumed at the end of the cultivation period. The production of off-flavours was recorded as the absolute concentration of the product at the end of the cultivation. All of the data were recorded in a CSV format for the ExcelTM worksheet and uploaded to ClustVis web tool. The clustering analysis followed the instructions available online by using the default parameters. The principal component analysis (PCA) used the singular value decomposition (SVD) method for single imputation, with unit variance scaling applied to rows, and data clustering based on correlations.

## 3. Results

### 3.1. Molecular Characterization of Yeast Strains

In the first screening, the yeast DNA was submitted to amplification of the ITS1-5.8S-ITS2 rRNA region and the results showed an amplicon of approx. 485 bp for all the yeast isolates (Figure 1). This specific species band pattern corresponded to what was reported for the specified *B. bruxellensis* [16]. After this, intraspecific characterization by RAPD using three OPA primers was performed and the resulting amplification profiles of each primer were combined to produce the strain-specific fingerprinting for each of the 16 isolates. The total number of bands obtained and the polymorphic bands were used to calculate the cophenetic correlation coefficient for the distance correlation defined by the binary indicator matrix (Table 1). With regard to this, the group with the highest coefficient is expected to be the best for describing the natural grouping of the matrix that is entered [20]. When the coefficient of cophenetic correlation is greater than 0.9, it can be interpreted as a very good fit, that is, there is a clear hierarchical structure between the objects, while values between 0.8 and 0.9 are considered to be good. On the other hand, values less than 0.8 or 0.7 are poor or very poor and suggest there is a clear disparity between the similarities and/or initial dissimilarities and those resulting from the graphic representation. This means that the data displayed in Table 1 are in an optimal range and show that the primers used were able to represent the similarity and/or dissimilarity between the RAPD patterns.

A cluster analysis of each isolate studied was conducted to assess whether there are differences between the strains with regard to the origin of isolation, and this was based on the primer with the highest percentage of polymorphism (OPAD08). This involved relying on the Jaccard similarity coefficient to cluster the isolates by means of the unweighted pair group method with arithmetic mean (UPGMA). The resulting dendrogram showed the differentiation of 15 isolates of *B. bruxellensis* (Figure 2). In this case, L-2763 and L-2731, which were isolated from the same winemaking process, showed full phenetic similarity and were considered to be isolates from the same clonal origin. For practical purposes, only L-2731 was used for the forthcoming experiments and comprised what will hereafter be called yeast genetic strains.

### 3.2. Yeast Response to Inhibitors

Small-scale cultures were carried out with 15 strains in either the absence or presence of mSO_2_, *p*-coumaric acid or a combination of both. In addition to their genetic differences, the 15 strains also had a similar growth profile in the presence of inhibitors, which confirms the degree of strain-specific sensitivity within the *B. bruxellensis* species (Figure 3). The first features that should be noted are the basic physiological characteristics of the wine isolates of *B. bruxellensis* in synthetic medium, which is designated the reference condition (Table 2). The very long lag phase varied from 19.5 h to 92.6 h (mean = 46.7 h; median = 38.9 h) while the very low growth rate varied from 0.003 h^−1^ to 0.035 h^−1^ (mean = 0.013 h^−1^; median = 0.011 h^−1^). Four out of 15 strains (L−2472, L−2474, L-2570 and L-2755) were very sensitive to mSO_2_ and their growth was completely inhibited at 0.3 mg L^−1^ (Figure 3; Table 2). The strain L-2482 started growing in the presence of 0.6 mg L^−1^ of mSO_2_ but only after a very long lag phase 83.3 h at an extremely low specific growth rate of 0.004 h^−1^ (Figure 3; Table 2). For this reason, most of the strains showed an increase in the lag phase in the presence of mSO_2_ that corresponded to the metabolic adaptation to this inhibitor.

When observing the growth kinetics of the strains studied only in the presence of 100 mg L^−1^ of *p*-coumaric acid, all showed an increase in the adaptation phase, varying from 21,998 h to 101,563 h (mean = 57.6 h; median = 57.6 h). Of the 15 haplotypes, five isolates were the only ones that presented an increase in their specific growth rate (L 2474, L 2597, L 2676, L 2679 and L 2763), of which the isolated L 2679 stands out, increasing this parameter by 75% from 0.008 (h^−1^) to 0.014 (h^−1^).

### 3.3. Response of Yeast Strains to p-Coumaric Inhibitors Acting Simultaneously with mSO_2_

The combined effects of mSO_2_ and *p*-coumaric acid were tested to evaluate the effectiveness of mSO_2_ as a contaminant-controller when there is a significant concentration of *p*-coumaric acid in the grape must. Overall, the presence of *p*-coumaric in the medium increased the lag time and reduced the growth rate by 30% in both cases. Despite the expected dose-dependent inhibitory effects of increasing concentrations of mSO_2_, an unexpected and significant protective effect of *p*-coumaric acid was observed on the mSO_2_ toxicity. For example, at the mean value for the 15 strains, the cells grew in the presence of 0.6 mg L^−1^ mSO_2_ at 39% of the growth rate of the reference condition. This was increased to 77% of the reference growth rate when *p*-coumaric acid was also present in the medium. Similarly, the extension of the lag phase obtained by the mSO_2_ dosage was reduced by *p*-coumaric from 135 h to 113 h as a mean value. It should be noted that the L-2474, L-2570 and L-2755 strains showed no growth in the presence of mSO_2_ at 0.3 or 0.6 mg L^−1^, but their growth was restored when the medium was supplemented with *p*-coumaric acid (Table 2). This means that, for the first time, we can show an antagonistic interaction, in which *p*-coumaric acid decreases the toxic effect of SO_2_. Furthermore, it has important implications for the winemaking process, where the grape must is used with high concentrations of *p*-coumaric contaminated by *B. bruxellensis*.

### 3.4. Metabolic Products in the Yeast Cultures

In the light of the results discussed above, *p*-coumaric acid turns out to be significant not only because it produces off-flavours, but also on account of its protective action against the toxic effects of SO_2_. Thus, there was a need to test the capacity of the yeast strains to consume and metabolise this molecule. All the *B. bruxellensis* strains were cultivated in synthetic medium in the presence of *p*-coumaric acid (100 mg L^−1^) as a precursor of the biosynthesis of the off-flavour volatile phenols 4-vinylphenol (4-VP) and 4 ethylphenol (4-EP). These three molecules form a part of the same pathway in which *B. bruxellensis* first converts *p*-coumaric acid to 4-VP via phenylacrylic acid decarboxylase (*PAD*) and then 4-VP is reduced by vinyl phenol reductase (VPR) to 4-EP. The average rate of *p*-coumaric consumption among the strains was 63.9% of its initial concentration, ranging from 15.5% for the L-2742 strain to 100% in the cases of L-2570 and L-2679 (Figure 4A). With regard to their resulting metabolization, the production of 4-VP ranged from zero to 94 mg L^−1^, while the production of 4-EP ranged from zero to 78 mg L^−1^. Most of the strains converted 100% of the consumed *p*-coumaric acid to at least one of these phenyl-derivatives, with variations among the strains (Figure 4B). In most of the strains, the percentage of 4-VP that was released to the medium was higher than the percentage of 4-EP, which means that the VPR enzyme is only partially working in these strains. This might be the result of enzyme deficiency or a lack of reducing power (NAD(P)H) for the reductive reactions. On the other hand, L-2570 converted 100% of *p*-coumaric to 4-EP (Figure 4B). Interestingly, the L-2679 strain consumed 100% of the *p*-coumaric acid added to the medium without producing any of the phenyl derivatives. This means that there might be another metabolic route for *p*-coumaric metabolization that does not convert its carbon chain to either 4-VP or 4-EP. In the same area, it can also be concluded that 34% of the *p*-coumaric acid consumed by the L-2759 strain was metabolised by this alternative pathway (Figure 4B).

### 3.5. Cluster Analysis of the Brett Wine Strains

Multivariate analysis was conducted based on four physiological conditions: extension of lag phase, growth rate, *p*-coumaric acid consumption and the production of volatile phenols. A relative index was used for the first two parameters, which means the ratio between the absolute value for each culture in treatment and the absolute value of the reference condition. The principal component analysis showed that the first component was responsible for 34.9% of the total variation and the second component for 22.3% (Figure 5A). Most of the strains showed a physiological profile that fluctuated around the median value of all 15 strains, with the exception of L-2676 and L-2679 strains (Figure 5A). By analyzing the data shown in Table 2, these strains showed a growth rate that was higher in all inhibitor-supplemented media than in the reference condition. Hence, these strains can be classified as highly tolerant to both agents (SO_2_ and *p*-coumaric acid). Overall, these strains only converted 50% of *p*-coumaric acid to 4-EP (Figure 4B), which revealed that this is not connected to sulphite tolerance. In addition, data clustering revealed some interesting features in the interaction between physiological parameters and yeast biodiversity (Figure 5B). First, there was the clustering of the set of data regarding the growth rate in the presence of inhibitory agents, together with the capacity of the strains to consume *p*-coumaric acid. However, this did not correlate with their ability to convert this substrate to phenyl derivatives, since L-2679 consumed all the *p*-coumaric acid provided and did not lead to any of these products (Figure 4). In fact, the ability to produce 4-VP and 4-EP was not related to the growth rate or to the extension of the lag phase (Figure 5B), which suggests that it is an independent variable with regard to sulphite tolerance. Moreover, the clustering analysis showed that there is no geographical distribution of the yeast strains, which is evidence that the ecological and industrial conditions of the different regions did not exert enough selective pressure to influence the physiological characteristics of the native strains. This result is in accordance with the genetic fingerprinting analysis that showed a greater diversity among the strains.

## 4. Discussion

Sixteen strains of *B. bruxellensis* that were isolated from fermentation processes in various regions of Chile were used to assess their genetic diversity and physiological characteristics with regard to their tolerance to *p*-coumaric acid and metabisulphite. Phenetic discrimination of the strains was carried out by RAPD, using three primers of the OPA series (OPAE09, OPAD08 and OPAE12) and this showed a high degree of discrimination with 15 separate fingerprints. Previous studies have demonstrated the importance of the choice of primers and the need to employ the technique to obtain optimal results [21,22]. For example, Miot-Sertier employed a method of typing to distinguish between different isolates of *B. bruxellensis* by RAPDS, by using four different primers (including OPAE09). The results showed that there was low heterogeneity in each group, in addition to a limited number of specific patterns per strain [23]. In contrast, Mitrakul showed that the PCR profiles generated by the OPAE09 primer were sufficient to type the *B. bruxellensis* strains studied [21,24]; this demonstrates that conditions for this experiment must be strictly standardized to ensure good reproducibility and accurate profiles.

By grouping the isolates from the same valley, each belonged to a different group, which means that, although this technique allows the strains to be genetically differentiated, they cannot be grouped by region or province [25]. The yeast strains that were isolated from the same wine region belonged to different phenetic groups. Hence, it is clear that this technique was sufficiently robust for the individual discrimination of the Brett isolates, since it succeeded in separating those that are different and grouping those that represent the same biological entity. However, this technique cannot be used to predict the physiological characteristics of a given isolate. This fact was previously described by Godoy [26], who, when differentiating 12 strains of *B. bruxellensis* using RAPD, did not find any relationship between the patterns of the isolates and their growth rates or enzymatic activities. For example, the phenetic group formed by L-2472 and L-2476 also yielded similar physiological data with regard to tolerance to the inhibitors. On the other hand, L-2480 shared a similar response to L-2472 and L-2476 with regard to the effect of inhibitors on the growth rate and only showed 15% of phenetic similarity to these strains. A similar situation was observed for L-2570 and L-2759, which were closely related in physiology, but remained unrelated by RAPD.

The strains identified by RAPD were physiologically characterized, and their ability to produce volatile phenols was evaluated, in addition to their growth and survival under stressful conditions. In assessing the ability of the different phenetic groups to produce volatile phenols, the culture medium was supplemented with 100 mg L^−1^ of *p*-coumaric acid, since the aromatic compounds (4-VP and 4-EP) are produced as a result of the metabolization of this hydroxycinmic acid [7]. Fourteen out of fifteen strains were capable of producing 4-VP, which is evidence that they all have phenylacrylic acid decarboxylase activity [27]. Moreover, most of them produced 4-VP as the result of active vinylphenol reductase [26]. The exception was the L-2679 strain that consumed all the *p*-coumaric acid without converting it to phenyl derivatives. This strain also had a high tolerance to SO_2_. Further experiments should be carried out to disclose this alternative pathway of *p*-coumaric acid metabolization.

The concentrations of volatile phenols were heterogeneous in the set of strains tested, without being correlated with their place of origin. This feature indeed showed that the capacity of producing off-flavours does not seem to be linked to the ecological and industrial conditions of the winemaking processes, even when the strains are genetically close [28].

Additionally, the effect of *p*-coumaric acid on the fitness of the different strains was evaluated. This substrate is commonly found in grape musts, so that its influence on the yeast performance might be an important feature in yeast adaptation and yeast spoiling activity. All of the samples showed the same general trend—they experienced a delay in the adaptation phase. This is mainly due to the inhibitory effect of hydroxycinnamic acids, which are weak organic acids that alter the intracellular pH and cellular metabolism [7]. In *B. bruxellensis*, it was found that the H^+^-ATPase pump (homologous to Pma1p in *S. cerevisiae*) plays a key role in reducing the concentration of intracellular protons during its adaptation to *p*-coumaric acid [17], which can be observed in changes in its growth kinetics. However, the data so far obtained cannot establish a direct connection between the presence of *p*-coumaric acid, and yeast tolerance to sulphite, which has serious implications for cytoplasmic acidification and ATP demand.

In the wine industry, SO_2_ is one of the most widely used additives due to its antimicrobial and antioxidant properties [29]. In general, the concentration of mSO_2_ used to prevent the growth of *B. bruxellensis* ranges from 0.2 to 0.8 mg L^−1^ [30,31]. The kinetic response of the fifteen identified strains was evaluated in synthetic minimum (YNB) and synthetic media supplemented with 0.3 or 0.6 mg L^−1^ of mSO_2_. In the case of most of the isolates, the presence of SO_2_ led to an increase in the adaptation phase, observed as the extension of the lag phase, as its concentration in the medium increased. However, the values obtained for the specific growth rate showed variations between the strains tested, regardless of the concentration of SO_2_ added. Five out of 15 strains tested were susceptible to the lowest SO_2_ concentration (0.3 mg L^−1^). One of the most widely known mechanisms of SO_2_ tolerance involves the activity of the SSU1 sulphite pump [32,33], a member of the Tellurite-resistance/dicarboxylate transporter (TDT) family. Diversity in sulphite tolerance has been found in strains from different geographical areas and with varying physiological patterns of behaviour [34,35], which suggests that this characteristic can be determined by quantitative factors. This is very important information since it is strongly recommended that mSO_2_ should be used when less tolerant genotype groups are included in the process [36]. However, this cannot apply to processes contaminated with L-2676 or L-2679 clonal groups. The existence of allotriploidy and genotype diversity in *B. bruxellensis* has been linked to high tolerance to SO_2_ [31,32,33,34,35,36,37,38], and has been attributed to the selection that occurs when adding this antimicrobial to the wide range of fermented beverages. If allotriploidy is the key factor in these two strains, this is a matter that requires further investigation.

The final quality of the wine results from a complex combination of factors, and these may be influenced by the type of hydroxycinnamic acid present in the grape must and the presence of residual SO_2_ and spoiling *B. bruxellensis* cells [39]. It is thus of great importance to evaluate the combined effect of these antimicrobial compounds on the fitness of the spoiling yeast. According to the results of Table 2 and Figure 4, tolerance to mSO_2_ and consumption of *p*-coumaric acid, including its conversion to volatile phenols, depend on both the substrate and the strain [39]. Potassium metabisulphite, which is one of the most widely used microbial contamination controllers during wine fermentation [40], can either kill the cells or simply impair cell growth. This last condition is described as a viable but not a cultivable state (VBNC) after sulphite stress [40]. In this scenario, it is essential to determine what kind of situations can interfere with this condition to ensure the full inhibitory activity of sulphite is retained, and to prevent its use at a higher concentration that can lead to a selection of tolerant strains and/or harm the sensory characteristics of the end product. To the best of our knowledge, this study has shown for the first time that *p*-coumaric acid can alleviate the inhibitory effect of sulphite for some strains, and thereby improving their fitness. Previous studies have revealed that the effect of different antimicrobial compounds against *Saccharomyces cerevisiae* present an unexpected variety of cellular responses when used in different concentrations and combinations [41], making it clearly complex to determine compound–compound interactions and the cellular response. In this sense, the improvement in yeast growth when exposed to both compounds could be due to more efficient responses by the cell, such as defense mechanisms (changes in metabolism, absorption and excretion of compounds, etc.) that allow it to survive more easily, compared to the mechanisms that are activated when the compounds are separated. We believe this effect should be the aim of future investigations in the field to ensure better control of spoiling yeast in winemaking processes.

## 5. Conclusions

This study confirmed that there is genetic diversity among yeast isolates of *B. bruxellensis* in different wine-producing regions of Chile, which is also represented by variability in the physiological characteristics regarding *p*-coumaric consumption and metabolization and sulphite tolerance. Despite the limitations with regard to the data that establishes a direct involvement of *p*-coumaric metabolization and sulphite tolerance, our multivariate analysis strongly suggested that this feature is also influenced by biological variables other than the SSU1 sulphite pump activity. In light of this, the L-2676 and L-2679 strains offer a suitable biological platform for carrying out in-depth studies on sulphite tolerance. In addition, the observed effects of *p*-coumaric in alleviating sulphite toxicity are of paramount importance. This is because it can provide a useful means of establishing the mechanisms that allow *B. bruxellensis* to tolerate different stressors, and thus forms the basis for future research into ways of determining the effects, occurrence and distribution of this industrially important yeast.

## Figures and Tables

**Figure 1 microorganisms-08-00557-f001:**
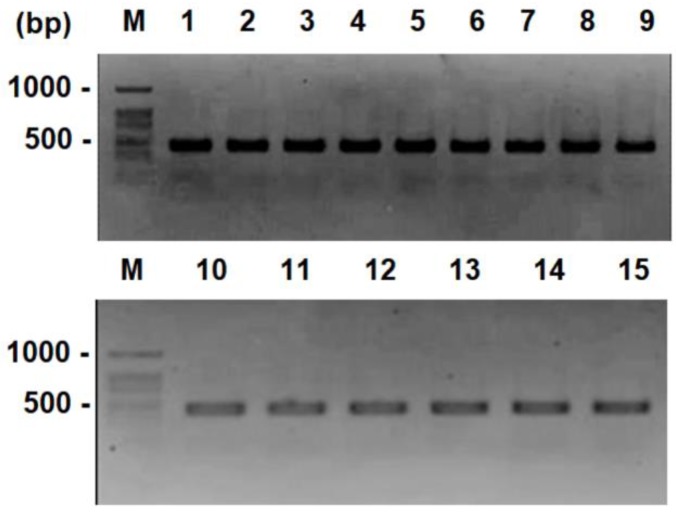
Amplification of the ITS1-5.8S-ITS2 segment from the DNA of *Brettanomyces bruxellensis* isolated from winemaking regions in Chile. Strains: L-2472 (lane 1), L-2474 (lane 2), L-2476 (lane 3), L-2480 (lane 4), L-2482 (lane 5), L-2570 (lane 6), L-2597 (lane 7), L-2676 (lane 8), L-2679 (lane 9), L-2690 (lane 10), L-2731 (lane 11), L-2742 (lane 12), L-2755 (lane 13), L-2759 (lane 14) and L-2763 (lane 15). 100 bp molecular weight was used (lanes M).

**Figure 2 microorganisms-08-00557-f002:**
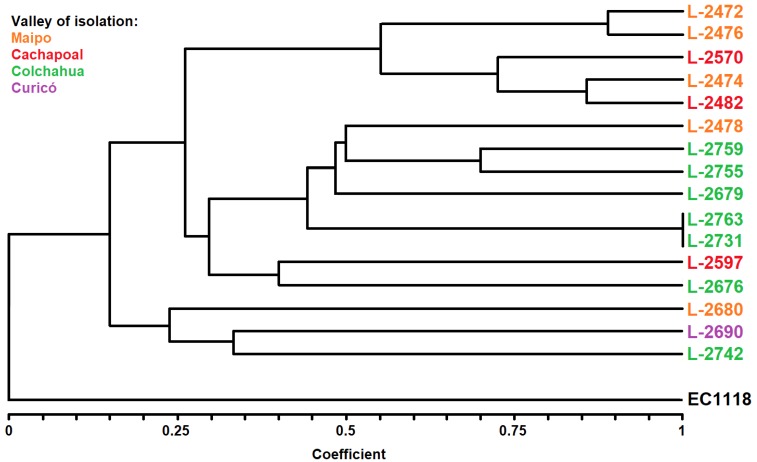
Dendrogram of genetic similarity among wine strains of *Brettanomyces bruxellensis* from Chile obtained by means of DNA fingerprinting patterns produced from RAPD technique using OPA series primers. The colors indicate the valley of origin of each strain.

**Figure 3 microorganisms-08-00557-f003:**
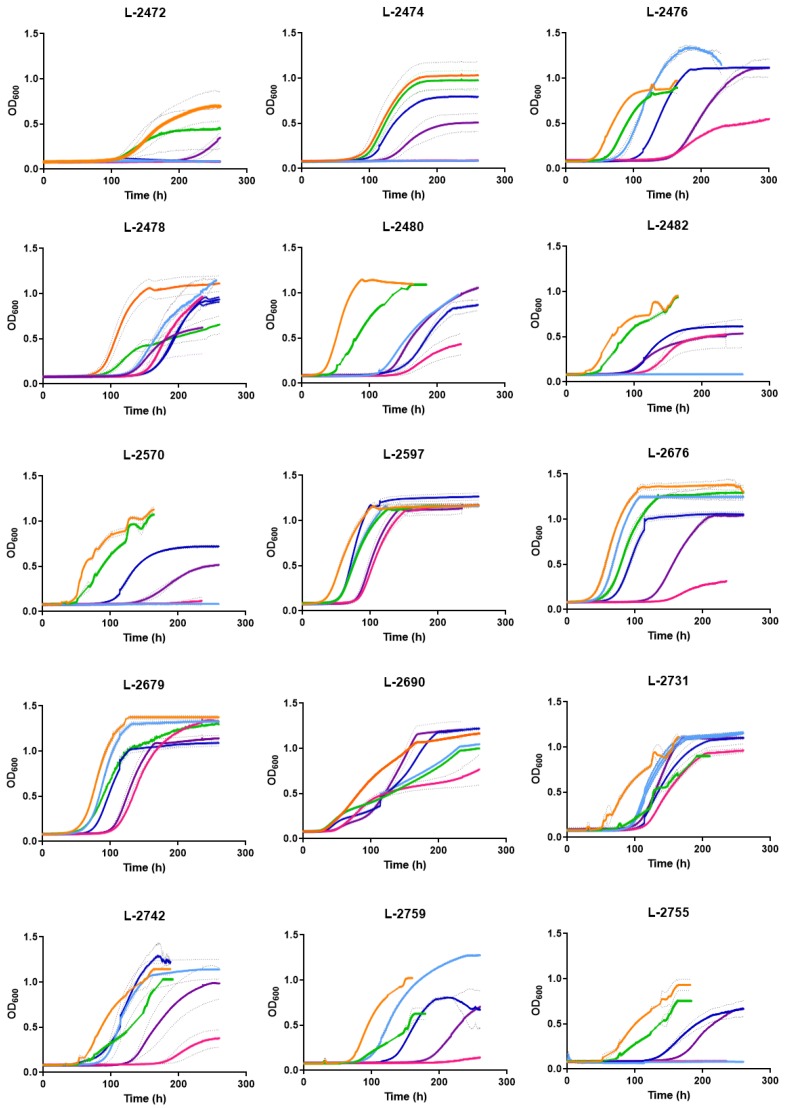
Growth curves of wine strains of *Brettanomyces bruxellensis* from Chile in synthetic (YNB) medium (orange curves),synthetic media supplemented with *p*-coumaric acid (100 mg L^−1^) (green curves), SO_2_ (0.3 mg L^−1^) (light blue curves), SO_2_ (0.6 mg L^−1^) (Pink), SO_2_ (0.3 mg L^−1^) and *p*-coumaric acid (100 mg L^−1^) (blue curves) and SO_2_ (0.6 mg L^−1^) and *p*-coumaric acid (100 mg L^−1^) (purple curves).

**Figure 4 microorganisms-08-00557-f004:**
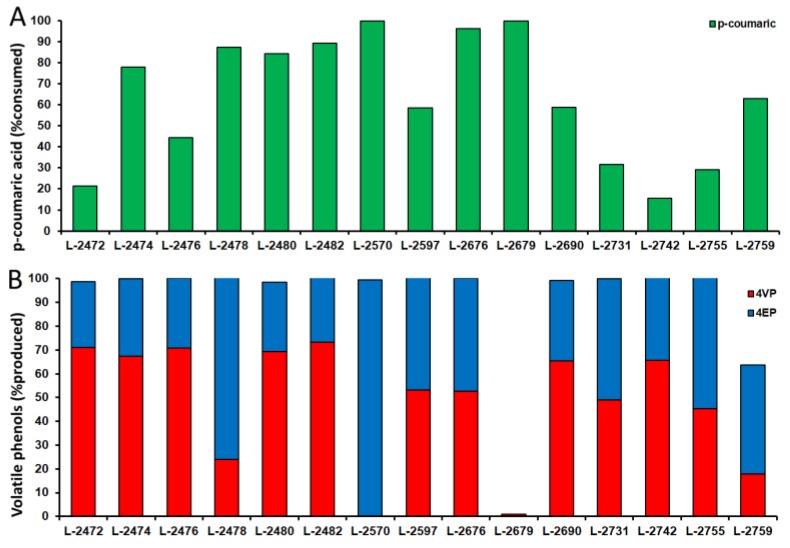
Relative consumption of *p*-coumaric acid (**A**) and relative production of volatile phenols 4-vinylphenol (4-VP) (red columns) and 4-ethylphenol (4-EP) (blue columns) (**B**) in cultures of wine strains of *Brettanomyces bruxellensis* from Chile.

**Figure 5 microorganisms-08-00557-f005:**
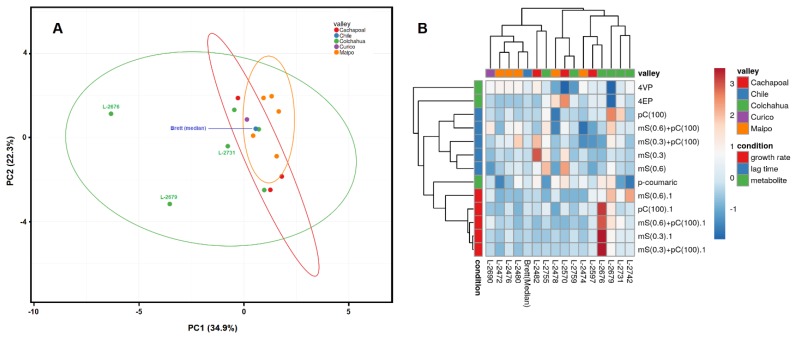
Multivariate analysis conducted on the basis of four physiological datasets of wine strains of *Brettanomyces bruxellensis* from Chile. Yeasts were cultivated in synthetic (YNB) medium supplemented with *p*-coumaric acid at 100 mg L^−1^ for evaluation of *p*-coumaric acid consumption (*p*-coumaric) (dataset 1) and production of 4-vinylphenol (4-VP) and 4-ethylphenol (4-EP) (dataset 2). Yeast cells were cultivated in synthetic media supplemented with *p*-coumaric acid 100 mg L^−1^ (pC(100)), SO_2_ 0.3 mg L^−1^ (mS(0.3)), SO_2_ 0.6 mg L^−1^(mS(0.6)), SO_2_ 0.3 mg L^−1^ and *p*-coumaric acid 100 mg L^−1^ (mS(0.3) + pC(100)) or SO_2_ 0.6 mg L^−1^ and *p*-coumaric acid 100 mg L^−1^ (mS(0.3) + pC(100)). Values were calculated for the extension of lag phase (dataset 3) and exponential growth rate (dataset 4) and reported as relative values to cultivation in medium without inhibitors. (**A**): Principal Component; (**B**): Clustering.

**Table 1 microorganisms-08-00557-t001:** Numbers and percentages of polymorphism of amplification products based on the different RAPD primers.

Primers	Total Number of Bands	Number of Polymorphic Bands	Polymorphism (%)	Cophenetic Correlation Coefficient
**OPAD08**	23	20	86.9	0.897
**OPAE09**	21	17	80.8	0.945
**OPAE12**	17	11	64.7	0.927
**Total bands**	61	48	77.5	

**Table 2 microorganisms-08-00557-t002:** Growth rate and extension of lag phase of *Brettanomyces bruxellensis* strains used in the study isolated from several wine-producing regions of Chile cultivated in synthetic medium in the absence or presence of *p*-coumaric acid and sulphite.

		No Inhibitor		*p*-Coumaric Acid (100 mg L^−1^)		SO2 (0.3 mg L^−1^)		SO2 (0.3 mg L^−1^) + *p*-Coumaric Acid (100 mg L^−1^)		SO2 (0.6 mg L^−1^)		SO2 (0.6 mg L^−1^) + *p*-Coumaric Acid (100 mg L^−1^)	
Strain	Valley	Lag (h)	µ (h^−1^)	Lag (h)	µ (h^−1^)	Lag (h)	µ (h^−1^)	Lag (h)	µ (h^−1^)	Lag (h)	µ (h^−1^)	Lag (h)	µ (h^−1^)
2472	Maipo	92.7 ± 2.5	0.005 ± 0.001	101.6 ± 4.6	0.002 ± 0.001	0	0	0	0	0	0	191.3 ± 2.499	0.003 ± 0.004
2474	Maipo	86.3 ± 1.6	0.013 ± 0.001	93.1 ± 3.5	0.014 ± 0.004	0	0	95.8± 5.648	0.010 ± 0.005	0	0	86.3 ± 3.22	0.003 ± 0.001
2476	Maipo	39.0 ± 1.3	0.016 ± 0.002	54.8 ± 2.6	0.013 ± 0.003	87.2 ± 3.0	0.022 ± 0.004	111.4 ± 8.562	0.017 ± 0.004	135.6 ± 4.9	0.002 ± 0.001	139.6 ± 4.441	0.011 ± 0.003
2478	Maipo	84.2 ± 0.7	0.018 ± 0.001	44.0 ± 1.7	0.003 ± 0.001	125.4 ± 5.6	0.009 ± 0.002	159.5 ± 3.885	0.011 ± 0.002	148.2 ± 3.7	0.011 ± 0.003	108.4 ± 2.554	0.005 ± 0.001
2680	Maipo	31.6 ± 2.2	0.035 ± 0.002	35.3 ± 2.4	0.010 ± 0.002	96.4 ± 4.9	0.010 ± 0.003	138.0 ± 2.445	0.008 ± 0.003	123.9 ± 4.2	0.003 ± 0.001	121.8 ± 2.645	0.009 ± 0.002
2482	Cachapoal	18.5 ± 1.7	0.011 ± 0.004	22.0 ± 3.6	0.008 ± 0.002	0	0	85.4 ± 4.533	0.007 ± 0.003	86.3 ± 2.5	0.004 ± 0.001	42.3 ± 3.441	0.003 ± 0.004
2570	Cachapoal	31.5 ± 1.4	0.014 ± 0.006	38.4 ± 3.7	0.014 ± 0.004	0	0	75.6 ± 2.114	0.009 ± 0.001	0	0	114.0 ± 2.566	0.003 ± 0.006
2597	Cachapoal	46.1 ± 2.4	0.021 ± 0.002	62.1 ± 3.1	0.019 ± 0.005	54.9 ± 4.3	0.022 ± 0.003	57.0 ± 2.159	0.029 ± 0.002	82.1 ± 4.3	0.019 ± 0.003	81.9 ± 1.823	0.022 ± 0.002
2676	Colchahua	49.7 ± 5.0	0.003 ± 0.002	67.9 ± 4.2	0.01 ± 0.002	51.8 ± 4.5	0.027 ± 0.004	71.8 ± 3.299	0.021 ± 0.002	139.4 ± 4.8	0.003 ± 0.002	125.4 ± 4.411	0.012 ± 0.003
2679	Colchahua	21.7 ± 3.2	0.008 ± 0.001	57.6 ± 2.9	0.014 ± 0.003	68.5 ± 2.4	0.026 ± 0.002	77.3 ± 4.558	0.021 ± 0.002	109.6 ± 2.6	0.016 ± 0.003	104.9 ± 2.956	0.019 ± 0.002
2731	Colchahua	35.3 ± 1.5	0.010 ± 0.003	79.0 ± 4.2	0.007 ± 0.001	94.5 ± 4.3	0.017 ± 0.002	105.9 ± 6.05	0.013 ± 0.002	112.0 ± 5.3	0.011 ± 0.003	107.3 ± 2.461	0.018 ± 0.004
2742	Colchahua	43.8 ± 2.4	0.011 ± 0.004	46.7 ± 4.2	0.009 ± 0.002	95.5 ± 2.3	0.018 ± 0.003	84.3 ± 3.24	0.016 ± 0.004	124.6 ± 4.6	0.024 ± 0.002	127.6 ± 3.224	0.010 ± 0.001
2755	Colchahua	34.0 ± 2.7	0.007 ± 0.001	59.3 ± 3.3	0.008 ± 0.001	0	0	111.9 ± 4.66	0.005 ± 0.002	0	0	145.8 ± 2.163	0.005 ± 0.001
2759	Colchahua	66.3 ± 3.6	0.014 ± 0.004	78.3 ± 2.6	0.007 ± 0.002	93.2 ± 4.3	0.013 ± 0.001	131.2 ± 3.685	0.013 ± 0.002	0	0	176.9 ± 4.998	0.007 ± 0.003
2690	Curicó	19.5 ± 2.8	0.009 ± 0.002	24.111 ± 2.4	0.004 ± 0.001	23.0 ± 2.5	0.005 ± 0.001	41.9 ± 4.627	0.007 ± 0.001	28.5 ± 1.4	0.006 ± 0.002	80.7 ± 1.437	0.012 ± 0.003

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
