# Peer review of "Biodiversity among *Brettanomyces bruxellensis* Strains Isolated from Different Wine Regions of Chile: Key Factors Revealed about Its Tolerance to Sulphite"

_microorganisms, 2020, doi:10.3390/microorganisms8040557_

Round 1
Reviewer 1 Report
The manuscript is relevant to winemaking industry because of the importance of B. bruxellensis as a spoilage microorganism and its resistance. The authors describe the effect of p-coumaric acid on the antimicrobial properties of SO2. This effect is really interesting, but the discussion of these results is a bit confusing. I recommend reviewing these statements. In addition, I have minor aspects to indicate before publication:
Abstract
Lines 19-21. Please rewrite the sentence.
Line 21: Indicate the meaning of the abbreviation RAPD.
Line 22: Indicate the meaning of “mSO2”.
Line 22: Please explain why is relevant to grow the microorganism in presence of p-coumaric acid and SO2.
Lines 24-26: In order to avoid misunderstanding I suggest to include that the effect of the presence of both compounds enhances the growth of the microorganism, why do you state that this is a synergistic effect? If p-coumaric acid mitigates the effect of SO2, they have an antagonistic effect, do not they? Please rewrite the sentence.
Introduction
Line 25. Remove a comma.
Lines 25-29. The objective of the study is confusing. I suggest: “The aim of this study was to investigate the genetic diversity, the physiological characteristics and the growth fitness in the presence of the combination of the antimicrobials SO2 and p-coumaric acid of 15 isolates of B. bruxellensis collected from fermentation processes in various regions of Chile. The evaluation of the yeast response to the inhibitors allowed understanding the complexity of the yeast resistance and their influence on the production of aromatic compounds”.
Results
Be consistent with the writing of p-coumaric acid (Italicize the “p”) throughout the manuscript.
Page 4, line 38. Italicize the name of the microorganism.
Page 4, line 40. Remove the spaces.
Page 4, line 46. Remove a comma.
Yeast response to inhibitors section. What is the effect of the incubation in presence of p-coumaric acid?
Response of yeast strains to p-coumaric inhibitors acting simultaneously with mSO2 section. The presence of p-coumaric acid produces a mitigating effect of the antimicrobial effect of SO2, is this an antagonistic effect?
Page 5, lines 25 and 28. Italicize the name of the microorganism.
Discussion
What can be the hypothesis of the mitigating effect of p-coumaric on the antimicrobial properties of SO2?
Author Response
Please, see the attachment.

Reviewer 2 Report
I have reviewed the manuscript entitled "Biodiversity among Brettanomyces bruxellensis strains isolated from different wine regions of Chile: key factors revealed about its tolerance to sulphite". It is an interesting study that was carried out to assess genetic diversity among yeast isolates of B. bruxellensis in different wine-producing regions of Chile and regarding p-coumaric metabolization and sulphite tolerance. This manuscript needs to be improved in terms of language and presentation. Attached you can find some examples of omissions and corrections. This manuscript can be accepted after serious methodological errors correction and text editing.
Examples that the manuscript language and presentation should seriously revised.
Page 2 line 6 “paramount importance”… check language
Page2 line 18 ….”of the beverage.” (correction: of wine)
Page 2 line 18-21 very high concentration of SO2 (correction: SO2 high concentration can lead to…).
Page 2 line 25 double comma
Page 2 line 32-37: Reference of activation method should be mentioned. Also activation temperature and time. Analytical origin of strains must provide. Temperature of incubation of petri?
Page 2 line 39-41 Reference of cultivation method? What about air flow and aerobic conditions? Water pH?
Page 3 line 18 Reference of activation method?
Page 3 line 24 Check spelling. (medium containing 100 mg/L p-coumaric acid)
Page 3 line 25 Exactly the concentration of SO2
Page 3 line 26 Why pH was adjusted to 3,5? Reference?
Page 3 line 29 Check spelling and provide reference of the method
Page 3 line 30 Agitation rpm?
Page 3 line 33-36 “When assessing the production of volatile phenols” spelling should be checked and language revised. Also provide exactly the temperature.
Page 3 line 37 The method should be referred as the origin (Ross, Beta and Arntfield)
Page 3 line 38-39 HPLC conditions must more detailed descripted.
Page 3 line 48 “All the data were recorded in a CSV format for the ExcelTM worksheet and uploaded to ClustVis web tool which is freely available online (https://biit.c 1 s.ut.ee/clustvis/)” Too much unnecessary informations.
Figure 3 must be redesigned
Author Response
Please, see the attachment
